# Positive and Negative Photoconductivity in Ir Nanofilm-Coated MoO_3_ Bias-Switching Photodetector

**DOI:** 10.3390/mi14101860

**Published:** 2023-09-28

**Authors:** Mohamed A. Basyooni-M. Kabatas, Redouane En-nadir, Khalid Rahmani, Yasin Ramazan Eker

**Affiliations:** 1Department of Precision and Microsystems Engineering, Delft University of Technology, Mekelweg 2, 2628 CD Delft, The Netherlands; 2Department of Nanotechnology and Advanced Materials, Graduate School of Applied and Natural Science, Selçuk University, Konya 42030, Türkiye; 3Laboratory of Solid-State Physics, Faculty of Sciences Dhar el Mahraz, University Sidi Mohammed Ben Abdellah, P.O. Box 1796, Atlas Fez 30 000, Morocco; redouane.en-nadir@usmba.ac.ma; 4Department of Physics, Ecole Normale Supérieure (ENS), Mohammed V University, P.O. Box 8007, Rabat, Morocco; khalid.rahmani@ens.um5.ac.ma; 5Department of Basic Sciences, Faculty of Engineering, Necmettin Erbakan University, Konya 42090, Türkiye; 6Science and Technology Research and Application Center (BITAM), Necmettin Erbakan University, Konya 42090, Türkiye

**Keywords:** molybdenum oxide, atomic layer deposition, sputtering deposition, urbach tail energy, low roughness optoelectronic devices

## Abstract

In this study, we delved into the influence of Ir nanofilm coating thickness on the optical and optoelectronic behavior of ultrathin MoO_3_ wafer-scale devices. Notably, the 4 nm Ir coating showed a negative Hall voltage and high carrier concentration of 1.524 × 10^19^ cm^−3^ with 0.19 nm roughness. Using the Kubelka–Munk model, we found that the bandgap decreased with increasing Ir thickness, consistent with Urbach tail energy suggesting a lower level of disorder. Regarding transient photocurrent behavior, all samples exhibited high stability under both dark and UV conditions. We also observed a positive photoconductivity at bias voltages of >0.5 V, while at 0 V bias voltage, the samples displayed a negative photoconductivity behavior. This unique aspect allowed us to explore self-powered negative photodetectors, showcasing fast response and recovery times of 0.36/0.42 s at 0 V. The intriguing negative photoresponse that we observed is linked to hole self-trapping/charge exciton and Joule heating effects.

## 1. Introduction

Photodetectors are an essential component of modern optoelectronics. Their widespread use encompasses various applications, such as optical communication, photoelectric imaging, and optical remote sensing. These devices play a crucial role in detecting and converting light signals into electrical signals, enabling the transmission of information and facilitating imaging processes in various fields [1,2]. Two-dimensional (2D) van der Waals (vdW) materials have become a fascinating option for photoelectrical conversion. This is because they exhibit a robust interaction between light and matter in the vertical direction, and they also offer excellent processing compatibility without the need to worry about lattice mismatching problems [3,4]. As an ordinary wide bandgap vdW semiconductor, α-molybdenum trioxide (α-MoO_3_) finds extensive use in optoelectronics [5], including gas sensing [6], solar cells [7], photodetectors [8], field-effect devices [9], and sensors [10,11]. The thin MoO_3_ layers exhibit a substantial k value, enabling excellent carrier mobility (>10^3^ cm^2^V^−1^s^−1^), a valuable characteristic for producing high-performance electronic devices [12]. Lately, researchers have discovered that α-MoO_3_ possesses exceptional hyperbolic properties, exhibiting tunable, in-plane anisotropic behavior, and demonstrating the ability to sustain ultralow-loss polaritons at mid-infrared (MIR) wavelengths [13].

MoO_3_ structure is used as a photodetector in many heterostructure devices. For instance, GaN/MoO_3–x_ nanorod array heterojunction has been demonstrated using a simple one-step physical vapor deposition method for 355 nm UV photodetector applications [14]. Jun Wang et al. fabricated a Bi_2_Se_3_/MoO_3_ thin film heterojunction for infrared photodetection applications with long-term environmental stability [15]. MoO_3_-based metal–semiconductor–metal photodetector device was used for 365 nm UV [16]. The device’s responsivity under UV light was 0.41 A/W at 7 V, with a rise time of 0.32 s and a decay time of 0.23 s. Another study conducted by Satyabrata Jit et al. fabricated an inverted MoO_3_ structure for optoelectronic applications to optimize the recombination current [17]. Another α-MoO_3_/Si photodetector for UV photodetection application had been fabricated using an e-beam evaporation method with a photoresponsivity of 63.3 mA/W [1]. Another work created a vertical MoO_3_/MoS_2_ heterojunction for photodetection and photovoltaic purposes [18]. The device design involved using MoS_2_ as the n-type region and its oxidation layer, MoO_3_, as the p-type region. This approach simplified the fabrication process of 2D vertical heterojunctions. Additionally, the device showed impressive photo response, boasting a photo-responsivity of 670 mA/W. The pyro-phototronic effect is successfully demonstrated in n-Si/MoO_3−x_ heterostructures [19]. Vertically grown 2D MoO_3−x_ microstructures with a centrosymmetric structure are synthesized for this purpose. The photodetector achieves impressive results, reaching a maximum responsivity of 4.4 mA/W and detectivity of 5.5 × 10^10^ Jones under photoelectric effects. To investigate the UV photodetector performance, MoO_3−x_ thin films are deposited onto SiO_2_/Si substrates using a sputtering technique that involves varying the oxygen partial pressure (pO_2_) at a substrate temperature of 200 °C [20]. Notably, increasing the pO_2_ leads to a reduction in the thin film’s thickness due to fewer atoms being ejected from the oxidized target surface during deposition with a responsivity of 1.537 mA/W, demonstrating its capability to detect UV light efficiently.

Iridium (Ir), a distinguished transition metal, has attracted considerable scientific attention due to its complex optical and electronic characteristics. One particularly significant aspect is its ability to exhibit plasmonic behavior, a phenomenon occurring within the UV to near-infrared spectral range, making it a promising candidate for various photonic and optoelectronic applications [21]. The foundation of Ir’s plasmonic response lies in the synchronized movement of free electrons at the intricate interface between the metal and dielectric substrates. This dynamic interaction results in a significantly enhanced local electromagnetic field, offering versatile functionality, especially in photodetection applications. Utilizing Ir’s plasmonic behavior for photodetection involves two primary mechanisms: the photoconductive and photovoltaic effects. The former involves harnessing the potent plasmonic field to enhance the photoconductivity of adjacent semiconductor materials. This phenomenon is practically realized by incorporating Ir as a plasmonic electrode alongside semiconductor counterparts like silicon or germanium. This orchestrated synergy leads to a noticeable increase in photocurrent, ultimately enhancing photodetection capabilities [22]. The intricate interplay between optical and electronic properties in metallic Ir plays a crucial role in defining its plasmonic features and photodetection capabilities. Optical characteristics are intricately linked to Ir’s electronic structure, significantly affecting its response to incident light. The electronic configuration, characterized by a partially filled d-band, contributes to notable reflectivity in the visible and near-infrared (NIR) spectrum [23]. While interband transitions in metallic Ir govern spectral interactions across the UV to visible range, the plasmonic response arises from distinct phenomena. The genesis of the plasmonic response in metallic Ir can be traced to the coherent oscillation of unbound electrons. This collective electron oscillation occurs at lower energy levels than interband transitions, resulting in a plasmonic range spanning from visible to NIR wavelengths. This spectral range has garnered substantial scientific interest due to its potential applications in various photonic and optoelectronic fields [24]. Therefore, the interplay of optical and electronic intricacies in metallic Ir offers a promising avenue for exploiting its plasmonic potential and enhancing its photodetection capabilities. Ir has emerged as a compelling choice for UV plasmonic applications, owing to its high bulk plasma frequency of 7.8 eV and a range of exceptional properties, including a high melting point, excellent electrical conductivity, and outstanding corrosion resistance [25]. The strategic incorporation of Ir into photoelectric devices is expected to provide increased stability and improved performance, highlighting its potential for transformative contributions to the field [26]. In a notable example of its capabilities, Ir has been employed in localized surface plasmon-enhanced UV photodetectors made from diamonds. Here, specific arrays of Ir nanoislands have been precisely engineered, and their impact on photoelectric performance has been thoroughly examined [23]. This not only demonstrates Ir’s role in advancing UV photodetection but also underscores its versatility in catalyzing scientific investigations aimed at enhancing optoelectronic functionalities. Additionally, a novel approach involves the incorporation of IrO_X_-co-catalyst-adorned plasmonic nanoparticles onto a semiconductor nanowire-array TiO_2_ electrode. This innovative design exhibits the dual ability to extend the lifetime of plasmon-induced charge carriers while simultaneously accelerating chemical reaction kinetics [27]. Such design innovations, built on Ir’s plasmonic properties, highlight its potential to facilitate multifaceted advancements, including prolonged charge carrier dynamics and faster chemical processes.

2D materials-based optoelectronic devices typically exhibit changes in current (photocurrent polarity) when exposed to light, and the magnitude of the photocurrent varies with light power. The observed photocurrent polarity depends on the dominant mechanisms operating in different devices, such as the photovoltaic effect, photo-thermoelectric effect, and bolometric effect [28,29]. Additionally, researchers have successfully modulated the photocurrent polarity by adjusting the gate voltage. This control over the gate voltage enables the manipulation of the competition among the photovoltaic effect, photo-thermoelectric effect, and bolometric effect, providing further flexibility and optimization in the performance of these devices. Recently noble metals such as Au on the surface of graphene induced a device with positive and negative photoconductivities [30]. The involved photoelectric measurements revealed that the device exhibited positive photocurrent when exposed to low laser power density and negative photocurrent when exposed to high laser power density. This behavior can be attributed to the underlying mechanisms affecting the device’s performance. At high laser power density, the mobility of graphene decreased due to the scattering effect caused by interactions between carriers and phonons (known as the bolometric effect). This effect was further enhanced by the presence of the Au nanofilm, leading to the generation of a local high-temperature field. These combined factors contributed to the observed negative photocurrent in the device under high laser power density illumination. The study based on the photo-thermoelectric effect discusses negative photoconductivity in flexible black phosphorus (BP) transistors constructed on a freestanding polyimide film [31]. When these transistors are illuminated by near-infrared laser light (λ = 830 nm), their on-state current is significantly suppressed leading to negative photoconductivity due to the strong photothermal effect. When the infrared light illuminates the device, it generates heat in the polyimide substrate. This heat, in turn, leads to enhanced phonon scattering and reduces carrier mobility, resulting in a negative photocurrent response. Another study investigated near-zero-biased copper selenide photodetectors based on negative photoconductivity induced by two mechanisms: photothermal effect and Joule heating [32]. 

Investigating the surface effect of ultrathin Ir films on the surface of MoO_3_ shows a negative photocurrent under zero bias voltage; however, it has a positive photoconductivity under a higher bias utilizing the impact of bias gate in controlling the photoconductivity of MoO_3_ thin films. For this purpose, this study investigates the experimental preparation and characterization of different Ir thin films. Meanwhile, electrical, structural, and optoelectronic properties were carried out to understand its performance as a switching photodetector device. 

## 2. Materials and Methods 

The first step involved cutting the n-type Si substrates into 1 cm^2^ sections to ensure consistent sample sizes. Subsequently, a cleaning procedure was carried out by immersing the substrates in acetone, isopropyl alcohol (IPA), and deionized water in an ultrasonic bath, with each solution being used for 20 min. The substrates were then dried in an oven at 60 °C and with the use of N_2_ gas. The surface of the substrates was activated using O_2_ plasma cleaner for 1 min before introducing them into the atomic layer deposition (ALD) chamber. For growing the MoO_3_ thin film, we utilized the ALD system (AT410 ALD system—ANRIC Technologies, Billerica, MA, USA). The Mo-based organo-metallic precursor material, Bis(t-butylimido)bis(dimethylamino)molybdenum (VI), from Strem Chemicals, Newburyport, MA, USA [33,34], was deposited on the substrate surface in an argon atmosphere (20 sccm—99.999% purity) at low pressure (~0.3 torrs). The amount of material deposited on the substrate surface depended on factors such as the open time of the valve between the growth chamber and the precursor material chamber, the temperature of the growth chamber, and the controlled heating of the precursor material chamber (around 80 °C). To prevent condensation between the growth chamber and the material chamber, the pipes were kept at 100 °C, and the outlet manifold was maintained at 105 °C. After deposition, the gases released from the substrate were discharged from the growth chamber with a flow rate of 10 sccm. In the next stage, O_3_ was carefully introduced into the growth chamber using an ozone generator (AT410 ALD system—ANRIC Technologies, Billerica, MA, USA), leading to the oxidation of the organo-metallic layer and the formation of a single layer of MoO_3_ thin film. Finally, the remaining gases were evacuated to complete the cycle.

To create an ultrathin layer of Ir, which was just a few nanometres thick, we used a Leica EM ACE600 US sputter coater. The process involved depositing this layer onto a high-vacuum film surface with a condition of 1 × 10^3^ mbar. To achieve this, we utilized Ar plasma with a depositing pressure of 5 × 10^−2^ mbar and applied a current of approximately 50 mA. The Ir target used had a width of 80 mm, and we monitored the thickness of the layer using a quartz thickness monitor. The sputtering rate for all the samples was 0.06 nm/s. To study the role of Ir, we deposited different thicknesses of 1, 2, and 4 nm, while keeping the MoO_3_ layer’s thickness constant at 600 pulses and deposited at 200 °C using ALD. Finally, a silver past was used as conductive electrodes connected to the probe station. 

In this study, various characterization techniques were employed to investigate the composition, morphology, topography, roughness, optical reflectance, thickness effect, and electric and optoelectronic properties of thin films. For surface morphology observation, we used the Zeiss Gemini 500 field-emission scanning electron microscopy (FESEM). To analyze the elemental composition of thin films, we utilized energy-dispersive X-ray spectroscopy (EDS), which can be attached to the FESEM. Atomic force microscopy (AFM) with the Park XE7 system was employed to examine topography and line profile spectrum in noncontact mode. The roughness values were measured using XEI 4.3.4 2016 data processing and analysis software. To characterize the electrical properties of the thin film sample, we used the SWIN Hall8800 Hall effect measurement system to measure carrier concentration and mobility. Parameters such as R_s_ (sheet resistance), R_ho_ (resistivity), V_H_ (Hall voltage), R_H_ (Hall coefficient), N_s_/P_s_ (sheet carrier concentration), and N/P (carrier concentration) were calculated. For electrical current–voltage and optoelectronics measurements, we employed a Sourcemeter and a four-probe system. Illumination with a 365 nm UV light lamp was used during the measurements. To measure the diffuse reflectance (R) spectra of the prepared samples in the range of 200–700 nm, we used UV-Vis-NIR Spectroscopy (UV-3600 Plus, Shimadzu, Kyoto, Japan). The reflectance spectra analysis provided insights into the electronic structure properties of the thin films. Finally, the FILMETRIC F20-UV thin film analyzer was utilized to determine the thickness of the thin films.

Throughout the experiment, the MoO_3_ layer thickness remained constant at 5 nm. However, for the Ir layers, the actual thicknesses were measured as 1.33 nm, 2.24 nm, and 4.38 nm, corresponding to samples 1 nm Ir/MoO_3_/Si, 2 nm Ir/MoO_3_/Si, and 4 nm Ir/MoO_3_/Si, respectively. Samples P1, P2, and P4 exhibited thicknesses of 1.33 nm Ir/MoO_3_/Si, 2.24 nm Ir/MoO_3_/Si, and 4.38 nm Ir/MoO_3_/Si, respectively.

## 3. Results and Discussion 

### 3.1. Surface Morphology

FESEM employs a high-energy electron beam to meticulously scan the surface of a sample, producing a detailed image that reveals its morphology. The findings for the P1, P2, and P4 samples are presented in Figure 1. The high-resolution morphology images illustrate that the P1 coating exhibits low homogeneity, displaying some agglomerations. However, as the thickness of the Ir coating increases to P2, the agglomerations become imperceptible, and the film exhibits high homogeneity across a large area. Furthermore, with a P4 sample, the grain size increases, leading to higher crystallinity.

### 3.2. EDS Elemental Analysis and Mapping

Energy-dispersive X-ray spectroscopy (EDS) analysis is a valuable method for studying the elemental composition of thin films. For the P1, P2, and P4 samples, it is observed that Mo is not present on the surface, confirming that the Mo signal will not be detected, as indicated in Table 1. When we examined the results in the absence of the Si peak, we found that the Mo signals were indeed zero, which was expected. However, the Ir concentrations were measured at 1.5%, 2.1%, and 5.6% for the respective samples. Furthermore, the O concentrations remained relatively constant at 1.3% for all three cases. Additionally, in the absence of Si peaks, the Ir concentrations were significantly higher at 62.8%, 71.2%, and 83.5%, while the O concentrations decreased to 37.1%, 28.8%, and 16.5%. We owe these insights to the capabilities of the EDS system.

The EDS mapping results have been presented in Figure 2. Notably, there are differences in the particle sizes between Ir and Mo due to their respective deposition methods. Ir particles, deposited using a sputtering system, appear larger than Mo particles, which are deposited using ALD. Generally, samples deposited by ALD tend to have smoother surfaces compared to those deposited by sputtering systems. In the case of the P1 sample, we can observe the presence of some particles, likely related to the Ir deposition process. On the other hand, for the P2 sample, we carefully selected smoother areas for the EDS analysis. Similarly, in the P4 sample, we noticed the presence of particles in specific smaller areas. Despite these variations, it is important to note that all the samples exhibit a high level of homogeneity overall.

### 3.3. Surface Topography and Roughness Analysis

In the field of surface metrology and topography, various parameters are used to characterize the surface features of a material. These parameters serve to quantify different aspects of the surface’s roughness and geometry. Some of these key parameters include average roughness (R_a_, in nm). It represents the arithmetic mean of surface heights within a specific area. The root mean square of the height values (R_q_, in nm) provides a measure of how much the height values deviate from the mean value, providing insight into the overall roughness. Height difference or peak-to-valley distance (R_pv_) measures the vertical distance between the highest peak and the lowest valley on the surface. Ten-point height (R_z_) calculates the average height of the five highest peaks and the five lowest valleys on the surface. Skewness (R_sk_) indicates the asymmetry of the surface features. Kurtosis (R_ku_) describes the sharpness or flatness of the surface features. These parameters are compiled in Table 2 for the P1, P2, and P4 samples based on the thickness of the Ir layer. In Figure 3, the corresponding roughness values of 0.199 nm, 0.235 nm, and 0.854 nm are calculated. Based on the FESEM images, the P1 sample exhibits a smooth surface with some small particles present. For the P2 sample, areas with smoother surfaces were selected, and the 400 KX FESEM images also revealed low roughness values. However, in the case of the P4 sample, the FESEM images indicate the presence of slightly larger particles, which aligns with the higher roughness values reported compared to the P1 and P2 samples.

### 3.4. Hall Effect and Carrier Concentrations

In this study, we conducted Hall effect measurements at room temperature, and the results are recorded in Table 3. The Hall effect measurements provided valuable information about the carrier concentrations of the samples. We observed that the Hall voltage (V_H_) was negative for all samples, which strongly suggests that these materials are of the n-type. This aligns well with the knowledge that MoO_3_ is naturally an n-type material. One interesting observation was the relationship between the carrier concentrations and the results of the Hall effect measurements. The carrier concentrations were found to be directly related to the measured Hall effect. As expected, since Ir is a highly conductive metallic element, we anticipated that the Hall effect system would reveal large carrier concentrations in these samples, which was indeed the case. An important factor that influenced the results was the thickness of the samples. We noticed that the sheet resistance (R_s_) decreased with an increase in sample thickness. This was primarily due to the high carrier concentration in the material. Notably, P1 had the highest sheet resistance (R_s_) at 1.028 × 10^6^ Ω/sq, indicating that it may possess higher electrical resistance compared to P2 and P4. The significance of the surface of the samples came into play during the Hall effect measurements. Since the Hall effect system collected signals from the near surface of the samples, the surface properties played a crucial role in determining the observed results. Additionally, when introducing a thicker Ir layer in the samples, we observed higher carrier concentrations. This trend was especially evident in the case of the P4 sample, which exhibited the highest carrier concentration among all the samples tested.

### 3.5. Reflectance, Bandgap, and Urbach Energy

The diffuse reflectance (R) spectrum can provide information about the absorption and scattering characteristics of thin films, as well as their bandgap energy. For example, a high reflectance in the UV region of the spectrum indicates that the film has a wide bandgap, while a low reflectance suggests a narrower bandgap. The R behavior shows that P4 has the lowest R followed by P2 and P1 samples as in Figure 4a. So, it can be confirmed the high response of the P4 sample is because this sample has the lowest R and consequently the highest absorption capability compared to the other samples, which might affect the photodetector performance.

Diffuse reflectance spectroscopy (DRS) is a widely used technique that utilizes a UV-visible spectrophotometer to investigate the optical properties of solid materials. Specifically, it is commonly employed to study the interaction of light with opaque and diffusely scattering substances [35,36,37,38]. To analyze the DRS results and estimate the optical bandgap energy of semiconductor materials, researchers often turn to the Kubelka–Munk (K-M) model [36,39]. The K-M model is a mathematical approach used to describe how light interacts with semi-infinite scattering media. It involves calculating a function denoted as F(R) or K/S, where ‘R’ represents the reflectance of the material, which is the fraction of incident light that is reflected. The K-M model is particularly valuable for characterizing materials that exhibit strong light scattering and absorption, such as pigments, coatings, and certain biological tissues. When it comes to estimating the optical bandgap energy of semiconductor materials, the Tauc plot is a useful method based on the absorption spectroscopy data obtained from DRS. However, in this context, the classical K-M model is often utilized due to its effectiveness in analyzing DRS results for such materials [38,40]. Figure 4b displays the bandgap results based on the K-M model. The data reveals a noticeable pattern where the bandgap reduces as the thickness of the Ir layer increases. Specifically, for the samples with 1 nm, 2 nm, and 4 nm Ir layers, the corresponding bandgap values are 2.30 eV, 2.12 eV, and 2.10 eV, respectively. This demonstrates a clear relationship between the thickness of the Ir layer and the bandgap energy of the material.

In certain crystalline materials, particularly those with disordered structures or defects, the energy bandgap does not have a sharp definition [41]. Instead, there is a gradual transition between allowed and forbidden energy states for electrons. Urbach energy quantifies this gradual absorption edge and represents the width of the tail of localized states near the band edges [42]. It is an important concept in understanding the gradual absorption behavior of materials, providing valuable information about their structural characteristics. It plays a significant role in the study of semiconductors and insulators [43]. To observe the Urbach tail, it usually plots the absorption coefficient (α) against the photon energy (hν) in what is called the Urbach region. The Urbach tail is characterized by an exponential decrease in the absorption coefficient as photon energy increases, beyond the main absorption edge corresponding to the bandgap. The mathematical representation [44,45] of the Urbach tail follows the Urbach rule:α(hν) = α_0_ × exp[(hν − E_0_)/E_u_]
where α(hν) represents the absorption coefficient at specific photon energy (hν), α_0_ is a constant related to the absorption intensity, E_0_ is the photon energy at the band edge or the onset of the primary absorption, and E_u_ is the Urbach energy, which signifies the characteristic energy width of the Urbach tail. A higher Urbach energy indicates more disorder or a greater number of localized states, while a lower Urbach energy suggests a more ordered and less defective material [46]. Based on this understanding, the results show that the 1 nm Ir layer has the highest Urbach energy, followed by the 2 nm Ir layer, and finally, the lowest value is associated with the 4 nm Ir layer. This indicates that the 4 nm Ir sample has a lower level of disorder or fewer defects compared to the other samples.

### 3.6. Refractive Index, Dielectric Constant, and Dielectric Loss

The complex refractive index is defined as follows:N(ω) = n + iK
where n is the refractive index, and K is the extinction coefficient. The extinction coefficient of thin films can be determined as follows:$$K=\frac{\alpha \lambda }{4\pi }$$
where α is the absorption coefficient of the thin film, and λ is the wavelength of the incident light. According to the extinction coefficient and the reflectance (R), the refractive index can be determined as follows:$$n=\frac{1+R}{1-R}+\sqrt{\frac{4R}{{\left(1-R\right)}^{2}}-K^{2}}$$

In addition, the complex electronic dielectric constant [47] is defined as follows:ε* = (n^2^ − K^2^) + 2inK
where the dielectric constant (real part) relates to the dispersion, while the dielectric loss (imaginary part) refers to the dissipative rate of the wave in the medium [48]. 

The refractive index as a function of energy is illustrated in Figure 5a. It appears that the 4 nm Ir sample exhibits the lowest behavior, while the 1 nm and 2 nm Ir samples demonstrate similar behaviors. Specifically, at a wavelength of 650 nm (1.9 eV), the refractive index values are as follows: the 4 nm Ir sample has a refractive index of 1.30, the 1 nm Ir sample has a refractive index of 1.53, and the 2 nm Ir sample has a refractive index of 1.60.

The dielectric constant (ε_real_) as a function of energy is shown in Figure 5b. When we look at the relationship between the refractive index and energy, we notice a similar trend in the dielectric constants of the different samples. The 4 nm Ir sample stands out by having the lowest dielectric constant behavior among them. Next in line is the 2 nm Ir sample, which shows a slightly higher dielectric constant compared to the 4 nm sample. Lastly, the 1 nm Ir sample exhibits the highest dielectric constant value among these three samples, suggesting a less pronounced response to the energy being applied. It means that the material represented by the 4 nm Ir sample does not store electrical energy as efficiently as the other samples. On the other hand, when we mention that the 1 nm Ir sample has the highest dielectric constant, it means that the material represented by the 1 nm Ir sample is better at storing electrical energy compared to the other samples.

In addition to the refractive index and dielectric constant, another important parameter, ε_imag_, representing the dielectric loss as a function of energy, is shown in Figure 5c. This parameter indicates the energy loss that occurs as the material responds to an electric field. Interestingly, it appears that the dielectric loss is higher than the dielectric constant for the same set of samples. Moreover, a similar trend to the dielectric constant is also observed in the dielectric loss for different thicknesses of Ir samples. Just like with the refractive index and dielectric constant, the 4 nm Ir sample demonstrates the lowest dielectric loss behavior among the tested samples. The consistent trend in the dielectric loss behavior with the thickness of the Iridium samples further emphasizes the influence of thickness on the material’s electrical properties. The 4 nm Ir sample, once again, stands out as having the lowest dielectric loss behavior, suggesting it may be the most suitable among the tested samples for applications that demand reduced energy dissipation such as self-powered photodetector devices. Overall, these findings are important for understanding the optical and electrical characteristics of Iridium samples with varying thicknesses. Such knowledge can guide the design and optimization of materials for specific applications in optics, electronics, and other relevant fields.

### 3.7. Current–Voltage Measurements

Figure 6 displays the electrical current–voltage behavior of three different Ir samples with thicknesses of 1 nm, 2 nm, and 4 nm under two different conditions: dark and UV illuminations. In the case of Ir/MoO_3_ (metal/semiconductor), it is reported that different metallic contacts such as Ag and Au metals can be used for collecting the output signals from this device [23]. Under dark conditions, as the thickness of the Ir layer increases, the electrical current also increases, and this behavior is more evident on a logarithmic scale. Specifically, the 4 nm Ir sample exhibits the highest current among the three samples. This implies that thicker Ir layers allow for a higher flow of electrical current, which may be due to changes in the material’s properties with varying thicknesses. Under UV illuminations, the presence of UV light results in a higher photocurrent for all three samples, regardless of their thickness. Interestingly, this increase in photocurrent is observed in both positive and negative voltages. Under negative voltages, the photocurrent reaches 150 µA for the 4 nm Ir sample, which is significantly higher compared to the photocurrent of 40 µA observed under positive voltages. This indicates that the 4 nm Ir sample is more responsive to the negative voltage when exposed to UV light, allowing for a substantial increase in photocurrent. These findings are further supported by the Hall effect measurements and bandgap values mentioned earlier. The 4 nm Ir sample exhibits higher carrier concentrations (the number of charge carriers in the material) and the smallest bandgap among the three samples. A smaller bandgap means that less energy is required to excite charge carriers, which may explain the higher photocurrent observed in the 4 nm Ir sample.

### 3.8. Transient Photocurrent Properties

In Figure 7, the transient current responses of P1, P2, and P4 thin films under both dark and UV illuminations are presented. These measurements are important to understand the stability of the sensor over time for stability assessments. The results show that the samples exhibit high stability at all different applied voltages, indicating the reliability of the sensor’s performance under various conditions. Interestingly, the thicker films of 4 nm Ir-coated Mo/Si demonstrate higher photocurrent behavior when exposed to UV illumination. Despite this increased photocurrent, these samples still maintain almost similar stability at different applied voltages. Specifically, the highest stable generated photocurrents under UV illumination at a 5 V bias voltage are 11.4 µA for the 1 nm Ir-coated sample, 18.9 µA for the 2 nm Ir-coated sample, and 44 µA for the 4 nm Ir-coated sample. These findings suggest that the 4 nm Ir-coated Mo/Si sample exhibits the most significant photocurrent response among the three thicknesses while maintaining stable performance under varying applied voltages. This information is valuable for optimizing the sensor’s design and functionality, particularly in applications where stability and sensitivity are critical factors.

### 3.9. ON–OFF Dynamics

The experiment focused on studying the light on–light off (ON–OFF) characteristics of Mo/Si samples coated with different Ir thicknesses of 1 nm, 2 nm, and 4 nm. It was carefully arranged and presented with precision over time to ensure a strong and reliable response to UV illuminations. To ensure the robustness of the photodetector’s performance, the UV light was allowed to rise and decay for 30 s, and this process was repeated multiple times. This meticulous approach aimed to verify the consistency and accuracy of the obtained data. Throughout the experiment, different bias voltages (0 V, 0.5 V, 1 V, 2 V, 3 V, 4 V, and 5 V) were applied, and the corresponding photocurrents for each set were recorded. The results were then carefully documented and graphed in Figure 8. The key findings revealed a notable trend: as the applied bias voltages increased, the generated photocurrents also increased correspondingly for all samples. However, the response to UV illuminations displayed a dependence on the thickness of the Ir layer. Specifically, as the Ir thickness increased, the photocurrent under UV light rose even higher, with the 4 nm Ir-coated sample showing the highest photocurrent generation among all three samples. Both the 2 nm and 4 nm Ir-coated samples exhibited stable photogenerated photocurrents, indicating their reliability for continuous operation. Notably, it was observed that a bias voltage of 0 V was insufficient to activate the photodetector, while just a slight increase to 0.5 V was enough to trigger its response. Throughout the entire experimental process, all samples demonstrated high stability and exhibited rapid response and decay behaviors, which are crucial attributes for efficient photodetectors. These results are supported by the previous findings of the IV, Hall measurements, and bandgap measurements which support that the 4 nm Ir sample is more interesting with higher photocurrents and stability.

### 3.10. Negative Photoconductivity at 0 V Bias

It is worth mentioning that all the samples displayed positive photoconductivity, meaning they produced a positive photocurrent when subjected to bias voltages ranging from 0.5 V to 5 V, as explained in the previous section. However, something intriguing was observed when the bias voltage was set to 0 V. Under this condition, the samples exhibited a negative photoconductivity behavior, as shown in Figure 9a. This indicates that at 0 V bias voltage, the photocurrent flow was in the opposite direction compared to the positive bias voltages. What is particularly interesting is that the magnitude of the negative photocurrent increased with the thickness of the Ir coating layer, with the highest value corresponding to the P4 sample. This suggests that the switching voltage may be a key factor influencing these negative photoresponses. For a better understanding of the dark current and the photocurrent effects in each sample, the I_ON_/I_OFF_ ratio is plotted in Figure 9b. It shows an increase with an increase in Ir thickness as confirmed by the ON–OFF figure.

The conclusion of the ON–OFF measurements of Ir-coated samples is in the importance of the deductible parameters from these curves such as response/recovery time, and responsivity. We see here that the Ir effect makes a high contribution to the photoconductivity and photostability of the MoO_3_/Si sensor at different applied voltages. As we have introduced, Ir is considered a promising material for thermo-plasmonic and surface plasmon applications. The main enhancement here in the UV photodetection is derived from the plasmonic effect of the ultrathin Ir layer [23]. We observe that the performance of the photodetector under different biases can be changed from positive to negative and vice versa as reported before [49,50]. It shows that the effect of different thicknesses of Ir is quite interesting for UV photonics even at a low bias voltage. The observed negative photoresponse [51] in the Ir-MoO_3_ films is related to the hole self-trapping/charge excitons [52], Joule effects, and photothermal heating effects [32]. Hole self-trapping/charge excitons: In a semiconductor, a hole can sometimes become trapped in its immediate vicinity due to its interaction with the lattice vibrations of the crystal. This is called hole self-trapping. When a photon is absorbed in the semiconductor material, it can generate an electron–hole pair. If the hole becomes trapped, it can form a charged exciton with the unpaired electron. The exciton can then dissociate, leading to the formation of a free electron and a trapped hole. Charge excitons can affect the efficiency of charge collection in a device, as trapped holes can reduce the photocurrent generated.

Joule and photothermal heating: In a device that has a finite resistance, a current flowing through it will generate heat due to the Joule effect. This heat can cause an increase in temperature, which can affect the performance of the device as reported for many materials such as metal oxides [53] and plasmonics [54,55,56]. Additionally, when light is absorbed in a semiconductor, it can lead to the generation of electron–hole pairs, which can also generate heat due to recombination processes. This is called photothermal heating. Both Joule and photothermal heating can lead to a reduction in the efficiency of the device, as they can affect the properties of the semiconductor material and the performance of the device. For instance, a study investigated near-zero-biased copper selenide photodetectors based on negative photoconductivity induced by two mechanisms: the photothermal effect and Joule heating [32]. These mechanisms contribute to the device’s exceptional sensitivity and efficient performance under low-bias conditions. The photothermal effect is achieved by illuminating the device, which generates heat and enhances phonon scattering. As a result, the carrier mobility decreases, leading to negative photoconductivity. Simultaneously, Joule heating plays a role in this phenomenon. Under low-bias conditions, the current passing through the device causes localized heating, further promoting negative photoconductivity.

### 3.11. Response and Recovery Time

Figure 10a presents the response and recovery time at zero applied voltage for three different thicknesses of Ir samples: 1 nm, 2 nm, and 4 nm. The response time refers to the time it takes for the sensor to detect and react to the incident light, while the recovery time indicates how quickly the sensor returns to its original state after the light source is removed. The standard definitions show that the time required for the photocurrent to rise from 10% to 90% is usually defined as the response/rise time, and similarly, the time required for the photocurrent to drop from 90% to 10% is defined as the recovery/fall time. One noticeable trend is that the response time decreases as the thickness of the Ir layer increases. In other words, the 4 nm Ir sample exhibits the fastest response time among the three samples. On the other hand, when it comes to the recovery time, both the 1 nm and 4 nm Ir samples show faster recovery times. However, what is intriguing is that the 4 nm Ir sample stands out with its fast response time and fast recovery time, making it an appealing candidate for applications that demand quick sensing capabilities. It is essential to highlight that the recovery time is slightly longer for all samples. This is attributed to the fact that halting the light and preventing further hole–electron pair generation naturally takes more time compared to the recombination process of existing hole–electron pairs.

Figure 10b displays the responsivity of the self-powered photodetector at 0 V for different thicknesses of Ir coatings. The responsivity calculations are derived from both the dark current (current when no light is present) and the illuminating current (current when exposed to light) for the three thicknesses of Ir layer: 1 nm, 2 nm, and 4 nm. As anticipated, the results reveal that the responsivity increases with the thickness of the Ir layer. Notably, the 4 nm Ir-coated sample exhibits the highest responsivity value of 0.8 A/W. This indicates that the photodetector with the 4 nm Ir layer is more sensitive to light compared to the other thicknesses.

## 4. Conclusions

In summary, this study explored the coating of Iridium (Ir) layers with varying thicknesses on MoO_3_/Si devices. The results showed that the thicker Ir samples exhibited higher carrier concentrations and negative Hall voltage, making them promising candidates for optoelectronic applications. We effectively analyzed the optical properties using diffuse reflectance spectroscopy and the Kubelka–Munk model, gaining valuable insights into the bandgap energy of the materials. Notably, the 4 nm Ir sample demonstrated the lowest dielectric loss, making it a suitable choice for energy-efficient applications like self-powered photodetectors. The electrical properties also varied with Ir thickness, with thicker samples displaying higher photocurrents, which can be advantageous for devices requiring greater current flow. We also observed a noteworthy effect with bias voltage: at 0 V, a negative photoconductivity was observed, while at higher voltages (0.5–5 V), a positive response was evident, showcasing the influence of bias voltage on switching effects in optoelectronic devices. Furthermore, the 4 nm Ir layer demonstrated fast response/recovery times and an impressive responsivity value of 0.8 A/W at 0 V, making it highly suitable for fast self-powered UV photodetectors. Overall, the current findings hold significant implications for utilizing Iridium-based materials in optoelectronics and sensors, taking into account their unique responses to light and electrical behavior based on thickness. These insights pave the way for enhancing the performance and efficiency of future optoelectronic devices.

## Figures and Tables

**Figure 1 micromachines-14-01860-f001:**
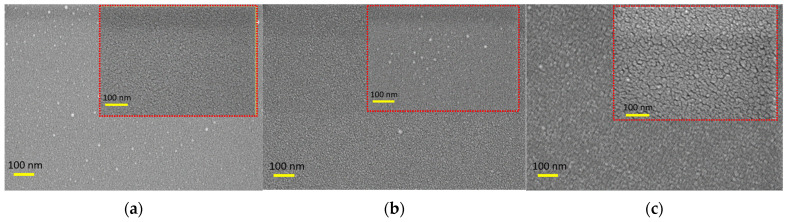
FESEM of (**a**) P1, (**b**) P2, and (**c**) P4 samples at 200 KX and 400 KX (inset) magnifications.

**Figure 2 micromachines-14-01860-f002:**
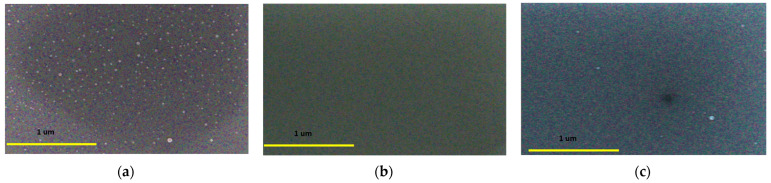
Ir/Mo/Si EDS layered images and elemental mapping distributions of O, Si, Ir, and Mo elements in (**a**) P1, (**b**) P2, and (**c**) P4 samples.

**Figure 3 micromachines-14-01860-f003:**
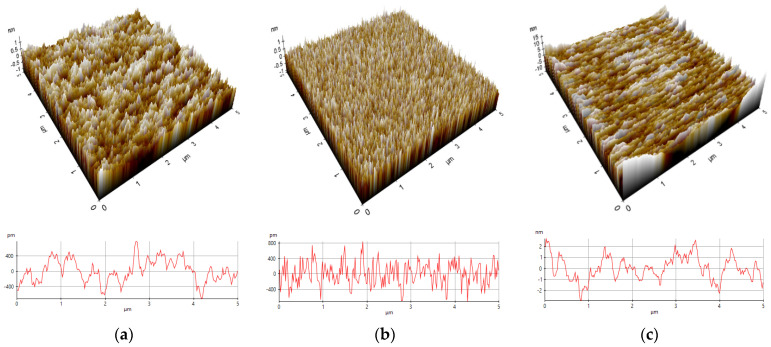
AFM 3D topography of Ir/Mo/Si and its horizontal line profile for (**a**) P1, (**b**) P2, and (**c**) P4 samples.

**Figure 4 micromachines-14-01860-f004:**
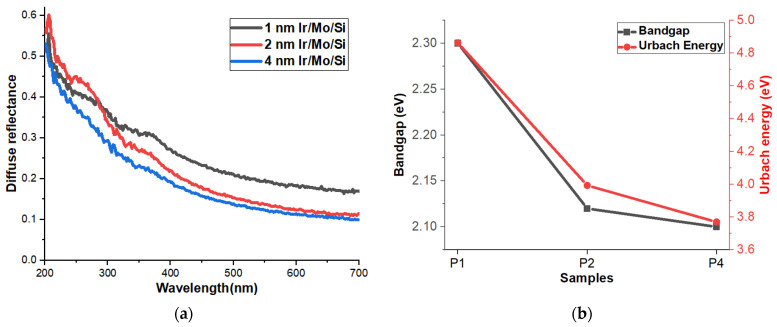
(**a**) Diffuse reflectance and (**b**) relation of the bandgap and Urbach energy of P1, P2, and P4 thin films.

**Figure 5 micromachines-14-01860-f005:**
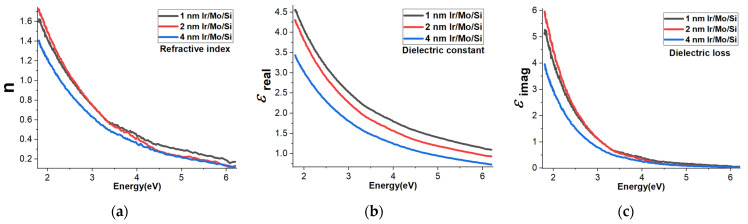
Variation of (**a**) refractive index n, (**b**) dielectric constant ε_real_ (**c**) and dielectric loss ε_imag_ on the energy of P1, P2, and P4 thin films, respectively.

**Figure 6 micromachines-14-01860-f006:**
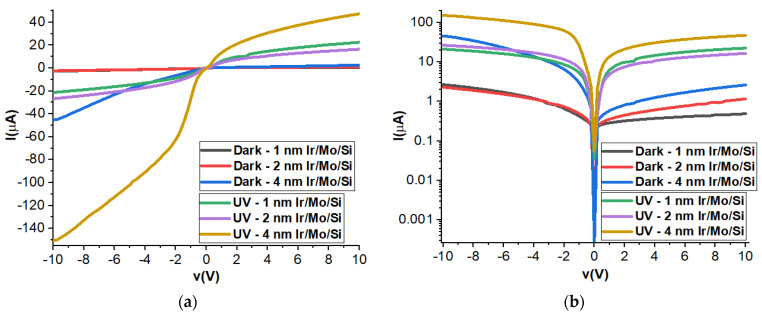
Electrical current–voltage behavior of P1, P2, and P4 thin films: (**a**) linear behavior and (**b**) logarithmic behavior under dark and UV illumination.

**Figure 7 micromachines-14-01860-f007:**
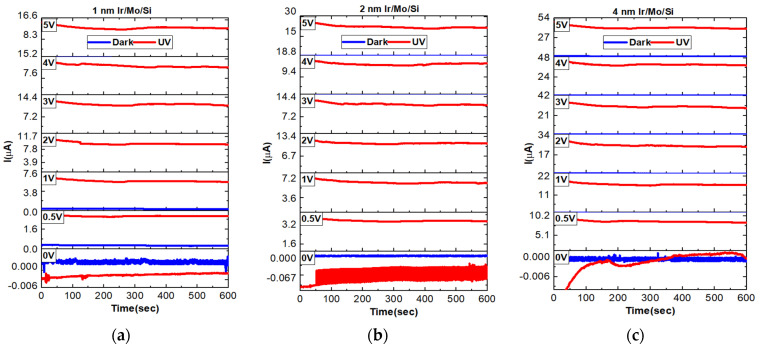
Transient photocurrent dynamics of (**a**) P1, (**b**) P2, and (**c**) P4 samples under different voltages of 0 V, 0.5 V, 1 V, 2 V, 3 V, 4 V, and 5 V.

**Figure 8 micromachines-14-01860-f008:**
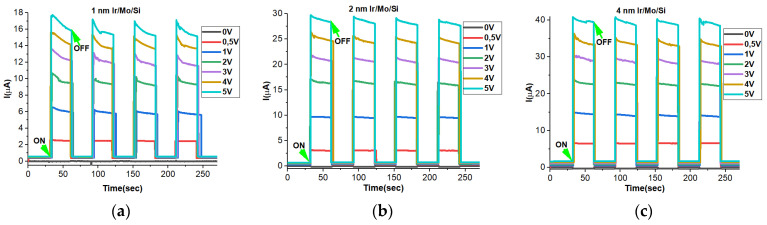
ON–OFF photoresponses of (**a**) P1, (**b**) P2, and (**c**) P4 samples at different applied voltages of 0 V, 0.5 V, 1 V, 2 V, 3 V, 4 V, and 5 V.

**Figure 9 micromachines-14-01860-f009:**
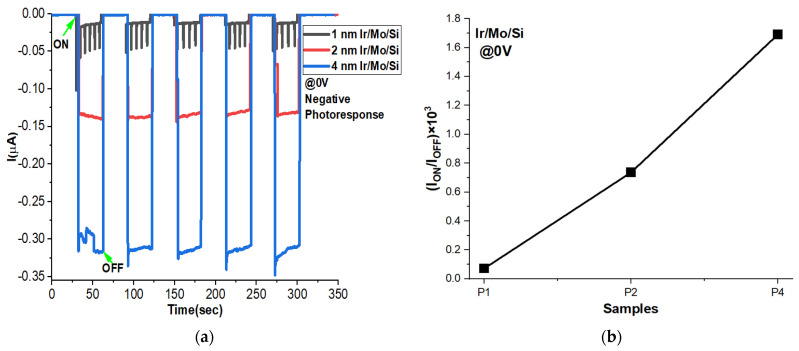
(**a**) ON–OFF photoresponse performances and (**b**) I_ON_/I_OFF_ ratios with time at zero applied voltage of P1, P2, and P4 thin films.

**Figure 10 micromachines-14-01860-f010:**
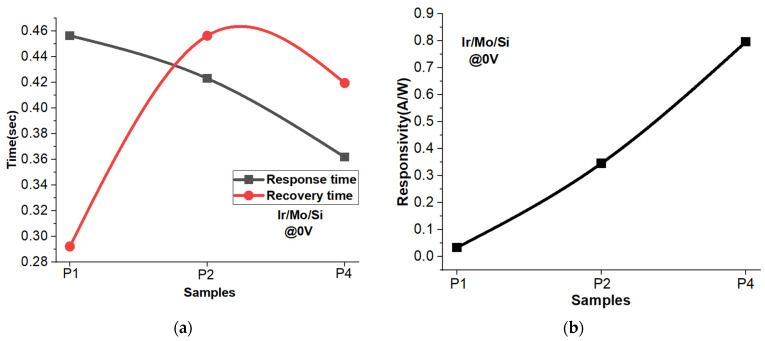
(**a**) Response/recovery time and (**b**) responsivity at zero applied voltage for P1, P2, and P4 samples.

**Table 1 micromachines-14-01860-t001:** Elemental composition amounts of the prepared samples in Wt %.

	Presence of Si				Absence of Si		
Samples	Si	O	Ir	Mo	O	Ir	Mo
P1	97.2	1.3	1.5	0.0	37.1	62.8	0.1
P2	96.7	1.2	2.1	0.0	28.8	71.2	0.0
P4	93.1	1.3	5.6	0.0	16.5	83.5	0.0

**Table 2 micromachines-14-01860-t002:** Roughness measurement of each group while the measurements were taken through a horizontal line.

Sample	Min (nm)	Max (nm)	Mid (nm)	R_pv_ (nm)	R_q_ (nm)	R_a_ (nm)	R_z_ (nm)	R_sk_	R_ku_
P1	−0.76	0.56	−0.10	1.33	0.25	0.19	1.09	0.22	3.05
P2	−0.72	0.84	0.06	1.56	0.29	0.23	1.37	0.02	2.82
P4	−2.94	2.69	−0.12	5.63	1.07	0.85	4.56	−0.07	2.81

**Table 3 micromachines-14-01860-t003:** Hall effect measurement at room temperature.

Sample	Rs (Ω/sq)	R_ho_ (/Ωcm)	V_H_ (V)	R_H_ (m^3^/C)	Type	N_s_ (/cm^2^)	N (/cm^3^)
P1	1.028 × 10^6^	7.449 × 10^−1^	−5.928 × 10^−3^	−1.984 × 10^−4^	n	−2.277 × 10^10^	−3.145 × 10^16^
P2	8.432 × 10^5^	5.338 × 10^−1^	−5.674 × 10^−3^	−4.981 × 10^−5^	n	−7.930 × 10^10^	−1.252 × 10^17^
P4	2.111 × 10^5^	2.584 × 10^−1^	−2.410 × 10^−5^	−4.092 × 10^−7^	n	−1.866 × 10^13^	−1.524 × 10^19^

## Data Availability

The data presented in this study are available on request from the corresponding authors.

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
