# Peer review of "Positive and Negative Photoconductivity in Ir Nanofilm-Coated MoO3 Bias-Switching Photodetector"

_micromachines, 2023, doi:10.3390/mi14101860_

Round 1

Reviewer 1 Report

1. This was primarily attributed to the high carrier concentration in the material. Why do 1 nm Ir/Mo/Si, 2 nm Ir/Mo/Si, and 4 nm Ir/Mo/Si exhibit different carrier concentrations?

2. On page 7, lines 223-235, we observed that the sheet resistance (Rs) decreased with increasing sample thickness, but 2 nm Ir/Mo/Si and 4 nm Ir/Mo/Si maintained their resistance (Rs). Furthermore, Rho, VH, and RH values were also consistent.

3. In Figure 6, 4 nm Ir/Mo/Si appears to have the highest photocurrent, but the dark current of 4 nm Ir/Mo/Si is also higher than that of 1 nm Ir/Mo/Si and 2 nm Ir/Mo/Si. To better understand the performance of these three structures, we kindly request the on/off rates for these devices. Additionally, we strongly recommend providing the corresponding on/off rates for Figure 8.

4. The unit of Rho in Table 3 should be (/Ω cm) instead of (/Ω-cm).

5. The labels on the left axis in Figure 7 (a), (b), and (c) are overcrowded and need adjustment. One possible solution is to shorten the horizontal axis and arrange the subgraphs for different voltages side by side.

6. It is important to note that in the field of photoelectric detection, the time required for the photocurrent to rise from 10% to 90% is usually defined as the rise time, and similarly, the time required for the photocurrent to drop from 90% to 10% is defined as the fall time. Please verify whether the manuscript's definitions (lines 462-464) and data (Figure 10 a) adhere to these standards.

7. The rise and recovery times (0.36/0.42 s at 0 V) is a very fast response. In factly, lots of photodetector shown microsecond  levels response time.

Author Response

Thank you for your support and efforts. 

Reviewer 2 Report

The Manuscript micromachines-2634498 Round 1

Dr. Mohamed A. Basyooni - M. Kabatas manuscript is devoted to the influence of Ir nanofilm coating thickness on the optical and optoelectronic behavior of ultrathin MoO3 wafer-scale devices. The work is practical in nature and is saturated with analytical methods for thin film studies. However, it requires revision and cannot be published in this form.

1.     The Keywords should not be repeated in the title and abstract.

2.     Line 29: Not clearly understand what authors want to describe. “GaN/MoO3–x has demonstrated using a simple one-step physical vapor deposition method”. It’s seems like they forget mention type of device which GaN/MoO3–x used.

3.     Line 102: “…negative photocurrent under zero bias voltage, however, a has a positive photoconductivity…”. Instead of “a” authors, perhaps, may have meant “it”.

4.     Line 110: IPA – it will be better if will write completely for the first time.

5.     Line 111: “…drying in an oven…” – which temperature?

6.     Line 113: ALD – the same remark as for the IPA.

7.     Line 130: Authors use the “E” for the measurement for Exponentiation, but I guess that better to use a more familiar form of 10x.

8.     Line 131: The same remark as for the line 130.

9.     It’s weird to introduce P1,P2,P4 samples and then still use 1 nm Ir/MoO3/Si, 2 nm

10.  Ir/MoO3/Si, and 4 nm Ir/MoO3/Si in the text and figs. And then, why authors use P1,P2,P4 – where is P3? But at the same time (line 161) they use P1, P2, and P3. Why? It seems like they use numbers according the thickness, but still not clearly why is P3 appears?

11.  Line 210: “These parameters have been compiled in Table 2 for the Ir/Mo/Si samples with 1, 2, and 4 nm Ir thicknesses.” How the thickness of Ir layers can be different in the same text? Why did the authors round the values without any explanation? “..for the Ir layers, the actual thicknesses were measured as 1.33 nm, 2.24 nm, and 4.38 nm…”. I suggest the following way: to avoid misunderstanding, use the notation P1, P2, P4 everywhere in the text.

12.  Line 237: :…resistance compared to P2 and P3.” Again

13.  Chapter 3.6 Refractive index, dielectric constant, and dielectric loss: prescribe formulas from a new line

14.  Fig.1: It is necessary to increase the labels on the figures. Especially in the minor FESEM images. Maybe try to increase both figs and captions.

15.  Fig.3: The images is too small, the graphics are even smaller. Authors need to use more clearer fonts for figs, make them bigger. But at the same time, the numbering of drawings should be done comparable to the font size in the text. It will look more carefully.

16.  Fig.6: Why are the graphs on a separate page?

17.  In general, paper was written in simple and understandable language. Introduction describes a lot of different authors works with variety of cases and background researches with MoO3–x at the same time, it does not overload with unnecessary information. It is written briefly and concisely. While reading the text, it seemed that different parts of the text were written by different people and in the final version of manuscript no one read it to check compliance. This worsens the holistic perception of the text. What about research part: many parameters were measured and analyzed. But the key study was to analyze the effect of the Ir layer on the properties of MoO3/Si devices. Accordingly, it is necessary to add data on the properties of Ir and its effect on similar structures to the Introduction. The authors' own arguments on this topic are not enough.

18.  In the introduction, the authors should definitely mention 10.3390/mi12060589 work on refractive index research and analysis, and 10.3390/ma16030993 work on the application of refractive index analysis in biosensorics.

19.     Line 110-111. The processing was carried out in a mixture of these agents (then indicate the composition) or sequentially in each of them? 20 minutes is the total processing time or in each of the agents (if consistent processing was carried out).

20.     Line 114:

ALD - it seems to me that all reduction should be introduced.

Anric Technologies (AT410) - not a cauldron manufacturer, his address ... I remember that we were required for detailed information (the same thing to line 124, which mentioned the ozone module of the same company).

21.     Line 115: Where does the reagent come from?

22.     Line 129: manufacturer Leica EM ACE600 US.

23.     Line 135: an united formatting of the degree of degrees when indicating the temperature in the lines for example, with lines 120 and 122.

24.     To what extent in the articles such a designation of the procedure is accepted: 5.0e-2? This is line 131. Another same style of designations in lines 19 and 130. At the same time, in line 43 the usual designation of the order of the size with an indicator of the degree in the upper index above the ten.

25.     Line 133. What is the Quartz Thickness Monitor? Is this some separate device? Then what kind of it? Where and by whom was done?

26.     Line 137: what a silver pasta, who did, what characteristics provides?

Again, this is a drying pasta like silver powder on a drying glue basis? Drying modes after application?

27.     Lines 151-152. What is the source of UV? Gas-discharge, LED, power, manufacturer, somehow ensured monochromaticity? What is she?

28.     Line 154: "The R Spectra Analysis Provided Insights ..." What is "R Spectra"?

29.     The lines of 159-161 showed that for the IR Layers, The Actual Thicknesses Weresured as 1.33 NM, 2.24 NM, and 4.38 NM, CorreSponding to Samples P1, P2, and P3, Respectively. Table 1 indicates rounded thicknesses 1, 2 and 4 nm. It seems to me wrong. You can conditionally call samples 1, 2 and 4 according to rounded values ​​of the thickness, but the table, which indicates the exact numerical data, must also indicate the thickness of the layer of iridium accurately, and not expect from the reader that he will keep the real thickness of the layers through the whole article , once read in lines 159-161.

30.     Fig 2. huge designations of figs (A), (b) and (c). Large rectangles seem to me completely non -informative. The sympathetic signature reads: "IR/MO/SI EDX Layered Images ..." What is EDX? In the article in line 175, this was called EDS and in the mention of the drawing in the test it says: "The eds mapping results have been presented in figure 2." You should dwell on one reduction.

Yes, and the signatures in the drawings are very small. The striking difference between the size of the characters in the text and in the drawings. Sometimes to complete inability in the figs.

31.     Lines 210 and 211 and wherever the thickness of the layer of iridium is indicated. Or call samples with conditional numbers on rounded thickness values, or indicate the exact measured thickness and not accustom the reader to the values that do not meet reality.

32.     Fig. 3. Not visible signatures on the axes. Let them redo it! The same applies to other drawings (most).

33.     Line 236. I do not like this style of representing the orders of the values: 2.4e+6.

34.     Fig. 4 (b): I did not understand how large -scale marks on the abscissa axis were set. This scale is clearly not linear. So what? The coordinates of points along the abscissa axis correspond to rounded thickness values, but this is wrong. If the schedule is built, then the coordinate corresponding to the thickness should be observed accurately, and the point is placed in accordance with the real value of the thickness. Moreover, the rounding is rather rude.

35.     Lines 256, 257: "Diffuse reflectance spectroscopy (DRS) is A widly used technique that uv-visible spectrophotometer to optical proterties aterials. " Great. And confirm with a reference to a number of articles or one monograph or a textbook?

36.     Lines 288, 301, 303 and 304. Is it possible to place formulas in the middle of the text?

37.     Fig. 5. The letter Epsilon in signatures to the ordinates axes in the drawings (b) and (C) is generally unreadable. I would not understand anything if I had not read the sympathetic signature.

38.     Line 364, syncrush signature to Fig. 6. "Electrical I-V Behavior ..." Here and in lines 150, 343 the voltage is indicated by a capital (large) letter V, and in the graphics of the lowercase V (small).

39.     Fig. 10. What kind of measurement units are indicated according to the abscissa axis ??? It used to be said at least that this is a thickness, but now in general a sample measured in nanometers? How is it at all? Well, and about the scale along the axis: what can be the scale of the sample?

Remove personal pronouns. Edit the style. 

Author Response

Thank you for your kind efforts and your time. 

Round 2

Reviewer 2 Report

The authors have worked hard and the work is much better. I recommend it for publication.

I recommend that the authors work on their style. In addition, they should get rid of personal pronouns that express excessive emotionality.